# Multi-omics approaches for comprehensive analysis and understanding of the immune response in the miniature pig breed

Devender Arora[1], Jong-Eun Park[1], Dajeong Lim[1], In-Cheol Cho[2], Kyung Soo Kang[3], Tae-Hun Kim[1], Woncheoul Park[1]*

1 Animal Genomics and Bioinformatics Division, National Institute of Animal Science, RDA, Wanju, Republic of Korea, 2 Subtropical Livestock Research Institute, National Institute of Animal Science, RDA, Jeju, Korea, 3 Department of Animal Sciences, Shingu College, Jungwon-gu, Seongnam-si, Korea

* wcpark1982@korea.kr

**Data Availability Statement:** All relevant data are within the paper and its Supporting Information files or can be retrieved underlying the results presented in the study on request from: https://

## Abstract

The porcine immune system has an important role in pre-clinical studies together with understanding the biological response mechanisms before entering into clinical trials. The size distribution of the Korean minipig is an important feature that make this breed ideal for biomedical research and safe practice in post clinical studies. The extremely tiny (ET) mini-pig serves as an excellent model for various biomedical research studies, but the comparatively frail and vulnerable immune response to the environment over its Large (L) size minipig breed leads to additional after born care. To overcome this pitfall, comparative analysis of the genomic regions under selection in the L type breed could provide a better understanding at the molecular level and lead to the development of an enhanced variety of ET type minipig. In this study, we utilized whole genome sequencing (WGS) to identify traces of artificial selection and integrated them with transcriptome data generated from blood samples to find strongly selected and differentially expressed genes of interest. We identified a total of 35 common genes among which 7 were differentially expressed and showed selective sweep in the L type over the ET type minipig breed. The stabilization of these genes were further confirmed using nucleotide diversity analysis, and these genes could serve as potential biomarkers for the development of a better variety of ET type pig breed.

## Introduction

Pre-clinical trials are the most effective measures taken to reduce the risk of any human calamities from a new treatment, and the animal model has an important role in serving such objectives. The significant information gathered from experimentation with lower mammals ultimately helps in the validation of hypothesis and is the true success of an experiment. The selection of suitable animals for the development of pre-clinical safe trials is a necessary prerequisite that would enable a strong foundation to pursue safe human-related trials [1, 2]. For these pre-clinical trials various animal models have been used, such as rodents, non-rodents, and non-human primates including minipigs which largely are considered the best animal model for the these studies due to various advantages [3–5]. Pigs as an animal model have been proven to have few ethical problems in

www.ncbi.nlm.nih.gov/geo/query/acc.cgi?acc=GSE184912.

**Funding:** This work was supported by Korea Post-Genome Project (Project title: Deciphering the reference genome and the discovery of trait-associated genes in Nanchukmacdon and mini pigs). Project No. PJ013343 of the National Institute of Animal Science, Rural Development Administration, Republic of Korea. This study was supported by 2020 the RDA Fellowship Program of National Institute of Animal Science, Rural Development Administration, Republic of Korea. This funding helped in successfully performing all the sample analysis and provided financial assistance to D.A. The funding bodies played no role in the design of the study and collection, analysis, and interpretation of data and in writing the manuscript".

**Competing interests:** The authors have declared that no competing interests exist.

studying pig organs for transplantation compared to pet animals (like the dogs) or large primates (like chimpanzees, orangutans, or gorillas), and the organ size is also an important advantage of minipigs that make the breed attractive for various biomedical research projects [6, 7]. Moreover, pigs as an experimental animal can be kept in a gnotobiotic/germfree chamber [8]. Such gnotobiotic pigs are an excellent model to examine kidney problems (hemolytic uremic syndrome) that occur after oral infection with enterohemorrhagic Escherichia coli [9, 10]. Furthermore, the immune system of minipigs is more than 80% similar to humans while the similarity of the mouse immune system to human is limited to 10%. Therefore, these advantages have popularized these pigs for feasible use in biomedical research [10, 11]. Minipigs are one of the key breeds anatomically and physiologically similar to humans that are used to understand various mechanisms and to evaluate the efficacy and safety of experimental therapies, drugs, and modalities in healthcare studies [12, 13]. Recent progress in genetic engineering also makes the minipig as ideal candidate to be used as a prospective organ donor for xenotransplantation in humans. There are different breeds of minipigs, e.g., Minnesota, Yucatan, Hanford, Mini-Lewe, and the widely used Göttingen minipigs [14, 15]. Among the different breeds of minipigs, only the Korean miniature-pig breed, Mini-Pig, registered with the United Nations, and Agricultural Food Organization (FAO) as a medical/laboratory species located in South Korea, and it is used for various biomedical research studies like xenografts, efficacy evaluation and biomaterial studies in different regions of the world. The Korean miniature pig varies from ET with an average body weight (18-26kg) to a L size minipig with an average body weight (37–85.6kg) range from birth to full maturation. Among the ET and L-type Minipig, the ET breed has been considered as one of the most suitable animal models, but due to a more prone immune system, the ET breed comes with the pitfall of extensive after birth care compared to the L type breed. Identification of genomic regions which undergo positive selection in one breed is a potent approach to delineate genes that help with adaptation to environmental factors and are responsible for the phenotype diversity.

In the last decade, many genome wide analyses with sound statistical approaches have been conducted to pin down significant results from the driven data [16, 17]. Using the existing WGS knowledge and understanding of molecular architecture, we oversaw the development of a breed with an enhanced trait value [18, 19]. These approaches already helped in the identification of different genomic regions with selection signals, suggesting the contribution of the region in influencing different traits related to phenotypic or genotypic composition [20, 21]. Similarly, with a better understanding of the genetic architecture and using advanced molecular breeding approaches these pitfalls can be overcome and a more stable ET breed can be developed with an L type immunogenicity response.

In the pursuit of identifying potential genes and their roles in different pathways, we performed WGS data analysis to distinguish selective sweep genes in the L-type over the ET type and RNA-seq analysis for their gene expression patterns. Here, we present an unbiased approach by integrating WGS and RNA-seq data and utilizing statistically established methods such as cross-population extended haplotype homozygosity (XP-EHH) [22], Integrated Haplotype Score (iHS) [23] and Z-scores of the pooled heterozygosity (ZHp) [24] statistics were used to detect the selection signatures from the ET and L type breeds followed by a comparative nucleotide diversity analysis on the identified genes of interest performed in the L vs ET type minipigs using vcftools to observe the stability in the region [25].

## Methods

The analysis was implemented to identify selective sweep genes in the minipig pig breed variety using re-sequencing data following the identification of differentially expressed genes by utilizing the RNA-seq data.

## Sampling and data collection

All the experimental procedures were verified and approved by the National Institute of Animal Science (NIAS) approval no: NIAS20181295 and carried out in consent with the ARRIVE guidelines [26, 27]. All the minipigs used in this study were male, and the average birth weight of the ET and L type was (0.18–0.3kg) and (0.5–0.7kg), respectively, and the average body weight at 12/24 months was 18/26kg and 37–85.76kg. The pigs were euthanized with an anesthetic injection given into the ear vein with an overdose of Alfaxan (0.7mg/kg), and blood samples were collected from post-harvested minipigs (N = 4) [28, 29]. Subsequently, the samples were stored in a sterile container and immediately frozen at −70°C until further analysis. RNA-seq data were generated for the minipigs (N = 4) with pair-end data after isolation of the blood samples using TRIzol (Invitrogen) and a RNeasy RNA Purification Kit with DNase treatment (Qiagen) following the manufacturers' instruction manual. One microliter of cleaned total RNA was used to check the RNA quality using BioAnalyzer with an RNA chip (RIN > 7 and 28S:18S ratio > 1.0). The library was constructed with random cDNA fragments and acquired adapter-fragments of the cDNA using the TruSeq Stranded Total RNA Sample Prep Kit (Illumina, San Diego, CA, USA) following the manufacturer's instructions. The constructed library was used to perform sequencing on the Illumina novaseq and paired-end reads were generated. and reported earlier [20, 30]. The selective sweep genes were identified from the data derived using WGS analysis or re-sequencing analysis performed by collecting the blood samples from the ET and L type Minipig breed. Similarly, differentially expressed genes were identified using applying the same blood sample for the RNA-seq analysis to obtain their role in the biological process.

## Sequence mapping and SNP calling

The raw reads were aligned with the reference genome of the pig (*Sscrofa11.1*) downloaded from the NCBI. SAMTOOLS was further used to clean low-quality map reads in the BAM files with permissive quality cutoffs [flag-sat–bS and–bF 4] [31]. To perform variant calling and snp/indels extraction, we used the Genome Analysis Toolkit 4.0 (GATK) pipeline based on best practices defined by the Broad Institute [32], and the Picard tool was used to filter potential PCR duplicates. Subsequently, the reference BAM file was indexed using SAMtools. Furthermore, the HaplotypeCaller, CombineGVCF and "SelectVariant" argument of GATK were used for the identification of single nucleotide polymorphism (SNPs). VariantFilteration was adopted from GATK to avoid possible false positive with the following parameters: SNPs with mapping quality (MQ) < 40.0, MQRankSum < − 12.5, ReadPosRankSum < − 8.0 and quality depth (unfiltered depth of non-reference samples; low scores are indicative of false positives and artifacts) < 2.0 were filtered [33]. Haplotype phasing and imputation of missing alleles for the entire set of swine populations were performed using BEAGLE version 4.1 [34]. After all the filtering processes, a total of 24,665,965 SNPs were retained and used for further analysis.

To perform differentially expressed gene analysis, the PE reads were first analyzed for the quality assessment using FastQC [35], and low-quality reads were removed using Trimmomatic tools [36] with parameters leading:3, trailing:3, slidingwindow:4:15, headcrop:13, and minlen:36 before proceeding to the sequence alignment. All quality-filtered PE reads were aligned to the *Sscrofa* genome (*Sscrofa11.1*) at the University of California Santa Cruz (UCSC) using Hisat2 [37, 38], and reads were counted using FeatureCount [39]. Finally, DESeq2 was used to identify differentially expressed genes [40].

**Selective sweep gene analysis.** To determine a genome wide pattern of positive selection using the whole SNP set identified from the ET and L type breeds, we first phased the SNPs data with a beagle and extracted each chromosome. Afterwards, we divided them into PopA

and PopB and applied three statistically established methods, XP-EHH [22], ZHp [24], and iHS [23], to detect the genome wide selective sweep regions. Here, each method based on different approaches such as the XP-EHH assesses haplotype differences between two populations. It is designed to detect alleles that have an increase in frequency to the point of fixation or near fixation in one of the two populations (A and B) being compared, by calculating the extended haplotype homozygosity (EHH) and log-ratio iHH between PopulationA and PopulationB as shown in Eq (1).

$$XP - EHH = \frac{\ln\left(\frac{iHH_A}{iHH_B}\right) - E\left[\ln\left(\frac{iHH_A}{iHH_B}\right)\right]}{SD\left[\ln\left(\frac{iHH_A}{iHH_B}\right)\right]} \tag{1}$$

Similarly, the iHS test is a program that identifies selected sweep genes by searching the locus where allele resides on a longer haplotype than the ancestral allele and compares the EHH between the derived and ancestral alleles as shown in Eq (2). This approach makes the method less affected by the demographic history and enabled us to identify incomplete sweeps, where the selected sweep is not fixed in the sample, and then, did a comparison between $iHH_A$ and $iHH_D$ denoted as ancestral and derived alleles [23].

$$iHS = \frac{\ln\left(\frac{iHH_A}{iHH_D}\right) - E\left[\ln\left(\frac{iHH_A}{iHH_D}\right)\right]}{SD\left[\ln\left(\frac{iHH_A}{iHH_D}\right)\right]} \tag{2}$$

To calculate ZHp we first obtained the expected heterozygosity (Hp) score at each position to scan the selection signals. The Hp values of individual SNPs were first calculated according to Eq (3) where ∑nMAJ and ∑nMIN represent the sums of the numbers of the major and minor alleles at each locus. Subsequently, to detect selection signals, the Hp values were then Z-transformed using Eq (4) [24, 41].

$$Hp = 2 \sum nMAJ \sum nMIN \bigg/ \left( \sum nMAJ + \sum nMIN \right)^2 \tag{3}$$

$$ZHp = (H_p - \mu H_p)/\sigma H_p \tag{4}$$

The genomic coordinates of the regions with a high XP-EHH, ZHp, and iHS score for the 10k window with a 10k bin size were computed using an in-house python script, and then, it was used as input data to fetch the gene_id information of the respective regions.

**Gene ontology analysis.** Lists of differentially expressed genes with p.adj ≤ 0.05 in the Minipig w.r.t. L vs ET type breed were compiled and submitted to DAVID v6.8 server [42] for functional annotation and enrichment analysis. Subsequently, list of degs visualized using Cytoscape program with string plugin [43, 44]. For each list, enriched Gene Ontology (GO) was performed for the 3 categories: Biological Processes, Molecular functions, and Cellular Compartments analysis. These terms were then clustered semantically using the ReviGO server [45]. Enriched functions throughout the whole transcriptome of the minipig with an elevated GO-term function and clustered lower-level GO-terms were visualized using treemap.

## Results

Blood samples from 4 minipig breeds (L and ET type), respectively, were collected, and we performed re-sequencing, which enabled us to obtain the complete genetic variation and to identify the genes potentially involved in genomic selection. Subsequently, RNA-seq analysis was performed which enabled us to identify differentially expressed genes.

## Population structure analysis

The principal component analysis (PCA) plot reveals the distribution of the two breeds in a two-dimensional view. The present PCA analysis from the WGS data was performed using the R package SNPRelate with 4 samples from each pig breed, and we obtained a clear separation of the respective breeds. Similarly, a clear separation was observed from the RNA-seq data after batch correction studies (Fig 1A and 1B).

## Genome wide artificial selection

Based on the high-quality SNPs, three tests were performed for the identification of positively selected genes in different genomic regions of the chromosomes of the L-type minipig breed. We identified positive selective sweep genes in the L-type minipig with $p \leq 0.05$ and respective scores of $\geq \pm 1.5$ and obtained 855 genes in XP-EHH, 3650 genes in iHS, and 2949 genes in the ZHp statistical analysis (S1 File).

## Differentially expressed gene analysis and common genes identification

The differential expression analysis was performed in R package DESEq2 after obtaining the gene expression count using featureCounts [39]. A cutoff value of the fold change $\geq \pm 1.5$ and adjusted p-value $\leq 0.05$ were selected to obtain differentially expressed genes (DEGs) between the respective breeds. The overall relationship differentially expressed pattern was further visualized by Volcano Plot (Fig 1C) [46], and to capture the information, representation of the gene interaction role was identified with Cytoscape [43] using the string database plugin [44] and presented as the network (Fig 2D). Next, common genes were identified among the iHS, XP-EHH, ZHp, and DEGs and the results were limited to 35 genes that were used for further analysis (Fig 1D, and S2 File). Amongst them, 7 were identified as differentially expressed and selective sweep genes in the L-type breed (Table 1).

**Gene ontology (GO) and gene regulatory network studies.** The information inferred from the existing literature reported a close relationship between human and pig organs [6, 47, 48]. KEGG pathway analysis of the identified commonly selective sweep genes from iHS, XP-EHH, and zHP was undertaken by filtering the data based on the score and a significant p-value $\leq 0.05$. The aims were to study the significance of the identified selective sweep genes in pigs and to comprehend the crucial functions shared by these genes in humans to extensively analyze and understand the molecular mechanisms shared by the species. Different genes were observed sharing common pathways such as the regulation of autophagy (ATG16L2, ATG7, and ATG13), Fc epsilon RI signaling pathway (MAP2K4, MAPK8, GAB2, and VAV2), Insulin resistance (MLXIPL, MAPK8, PRKCE, CREB3L1, and PTPRF), Axon guidance (ROBO2, EPHA5, DPYSL5, SRGAP1, and ROBO1), and cAMP signaling pathway (CAMK2B, MAPK8, CREB3L1, HTR4, VAV2, and RAPGEF4) (Table 2, and S1 Fig). Amongst them, the commonly enriched pathways were axon guidance (GO:0007411), synaptic vesicle endocytosis (GO:0048488), microtubule cytoskeleton organization (GO:0000226), negative regulation of cell migration (GO:0030336), protein localization to basolateral plasma membrane (GO:1903361), camera-type eye photoreceptor cell differentiation (GO:0060219), hippocampus development (GO:0021766), and vesicle-mediated transport (GO:0016192) in Biological process. Similarly, cell-cell junction (GO:0005911), synapse (GO:0045202), basolateral plasma membrane (GO:0016323), and axoneme (GO:0005930) were enriched in the cellular component and transferase activity, and transferring glycosyl groups (GO:0016757) was identified in the molecular function (S3 File).

Similarly, a separate analysis was performed for the enrichment analysis in positively and negatively expressed DEGs in the L-type minipig. We observed signaling pathways such as the

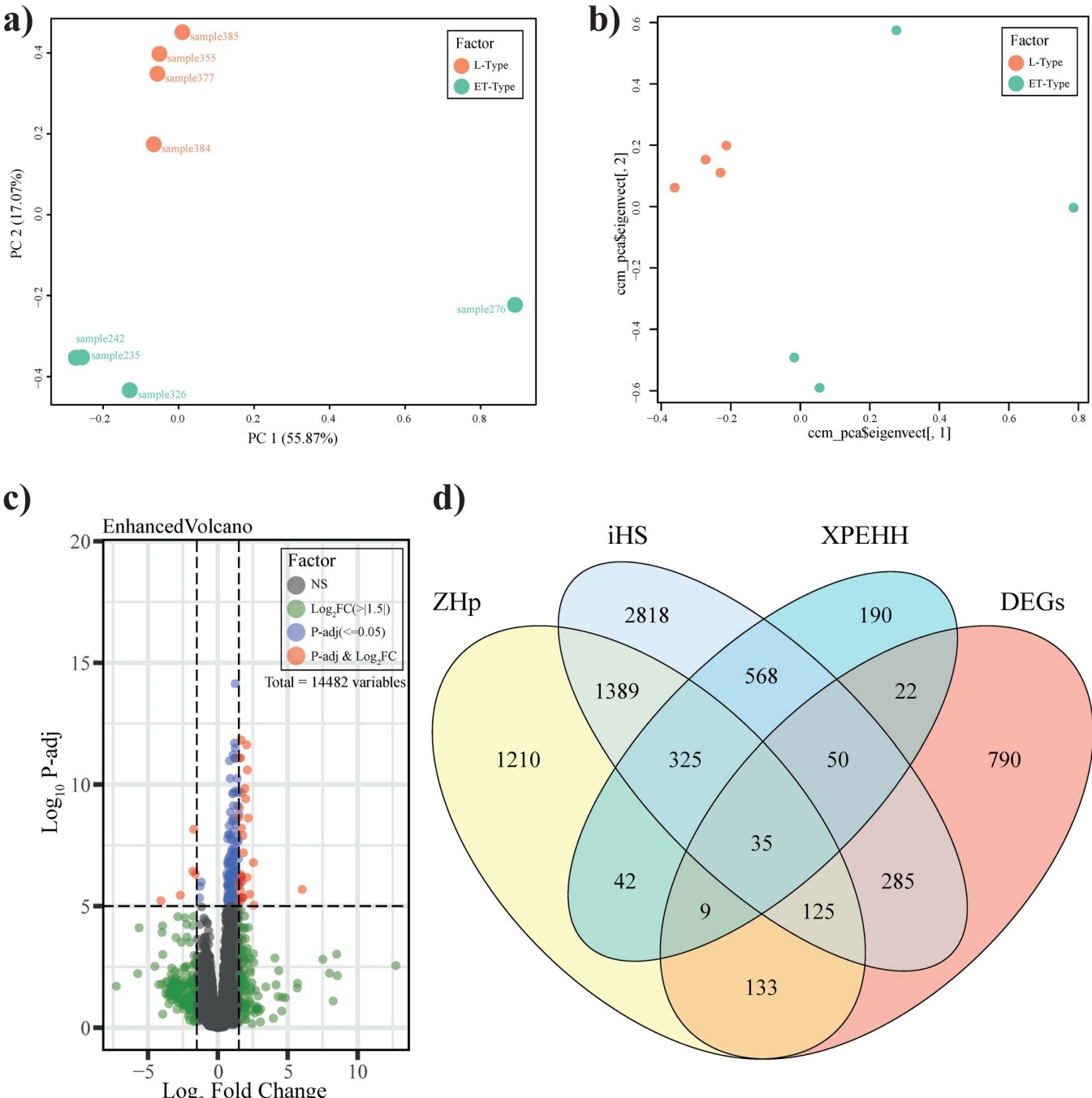

**Fig 1.** Principal component analysis: (a) & (b) representing the distribution of pig breed variety (L vs ET) in 2-d view. There a clear separation between the breeds can be visualized. (c) Differentially expressed genes visualization was performed using enhanced volcano plot with p.adj ≤ 0.05 and Log2FC ≥±1.5, here NS signifies non-significant genes. (d) Common genes in different condition viz. iHS, ZHp, XP-EHH, and DEGs were visualized using Venny, and 35 common genes were identified.

TNF signaling pathway, NF−kappa B signaling pathway, Rap1 signaling pathway, Neurotrophin signaling pathway, Cytokine−cytokine receptor interaction, and Toll−like receptor signaling pathway. Among them, the major pathways were enriched in the upregulated condition (Fig 2C). Likewise, the functional annotations of genes were tagged into three groups:

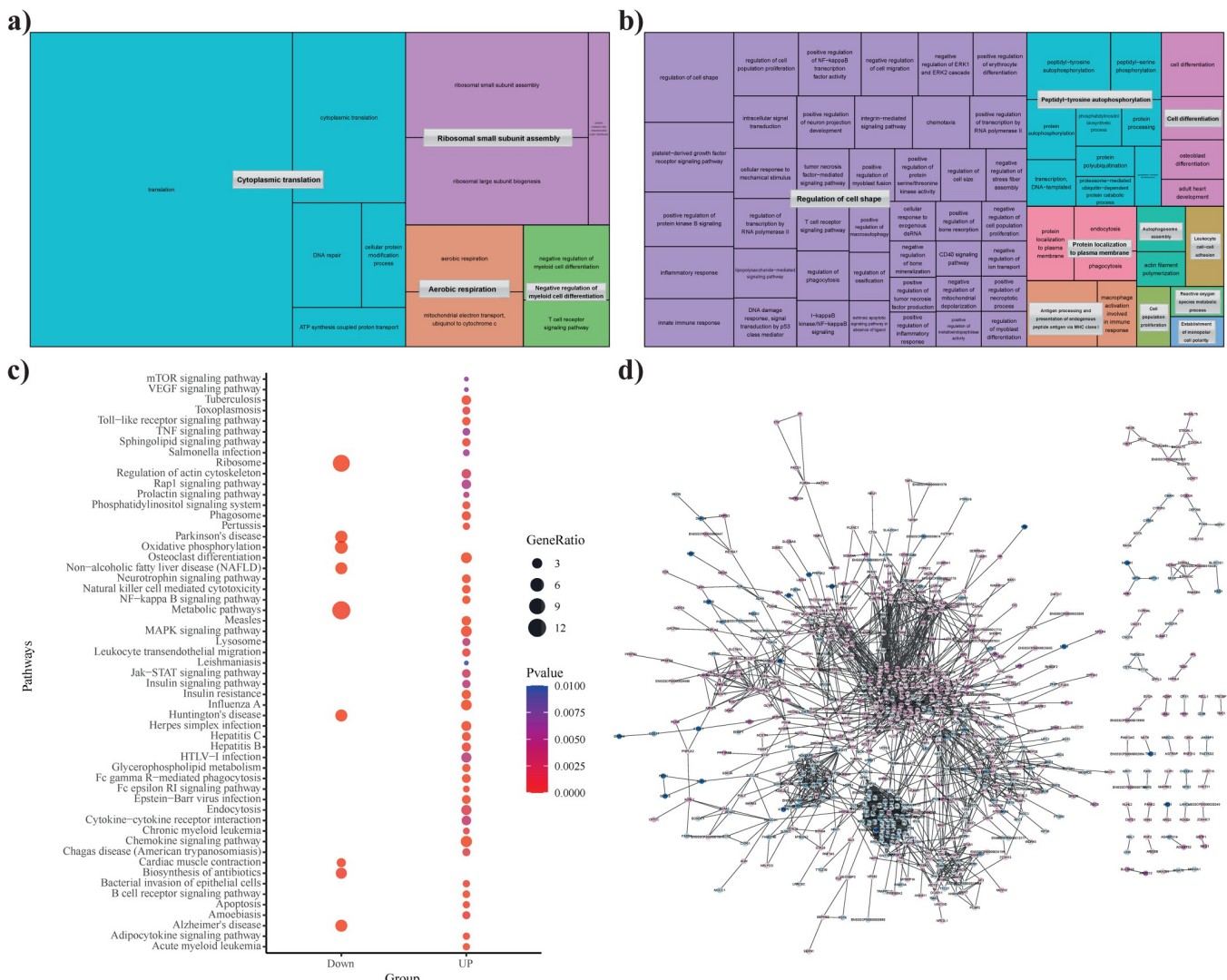

**Fig 2.** (a and b) Gene ontology study was done to identify the contribution and significance of differentially expressed upregulated and downregulated genes in minipig with p ≤ 0.05. (c) KEGG pathway enrichment analysis after functional annotation with p < 0.01. Enriched pathway in L-type minipig was performed by dot-plot analysis. (d) protein-protein interaction analysis was done to visualize the upregulated and downregulated genes. Here, Blue nodes represent the downregulated genes and purple nodes represent the upregulation of genes.

**Table 1. Identification of differentially expressed selective sweep genes.**

| Gene_Id | ENS_id | Chr | XP-EHH | iHS | ZHp | log2FC | FDR |
|---------|--------|-----|--------|-----|-----|--------|-----|
| *AIF1L* | ENSSSCG00000034178 | 1 | 2.649219181 | 2.6971365 | -2.5781349 | 1.7994906 | 0.02328175 |
| *DNAH9* | ENSSSCG00000018015 | 12 | 4.629307125 | 4.1900572 | -3.3738688 | 2.1705897 | 0.0000756 |
| *GABBR2* | ENSSSCG00000027558 | 1 | 3.922787712 | 5.2696931 | -2.5781349 | -2.523233 | 0.01536176 |
| *GRTP1* | ENSSSCG00000009554 | 11 | 3.34265521 | 3.69934 | -4.2762313 | 1.5207297 | 0.0000014 |
| *HBB* | ENSSSCG00000014725 | 9 | 3.111822546 | 3.0945901 | -1.866248 | -3.382391 | 0.02276081 |
| *HECW1* | ENSSSCG00000036443 | 18 | 3.32804312 | 5.2894497 | -3.963867 | 1.9930709 | 3.87E-10 |
| *HTR4* | ENSSSCG00000014428 | 2 | 2.788343178 | 2.2548874 | -3.9885142 | 1.7281431 | 0.00707621 |

**Table 2. Comparison of KEGG pathways enriched in Pig and Human for selective sweep genes.**

| SNO | KEGG Number | KEGG Pathway | Pig Genes | Human Genes |
|---|---|---|---|---|
| 1 | ssc04360: hsa04360 | Axon guidance | *ROBO2, EPHA5, DPYSL5, SRGAP1, ROBO1* | *ROBO2, EPHA5, DPYSL5, SRGAP3, SRGAP1, NTN1, EPHB1, ROBO1* |
| 2 | ssc04024: hsa04024 | cAMP signaling pathway | *CAMK2B, MAPK8, CREB3L1, HTR4, VAV2, RAPGEF4* | *CAMK2B, GABBR2, MAPK8, GRIN3A, CREB3L1, HTR4, VAV2, RAPGEF4* |
| 3 | ssc04664: hsa04664 | Fc epsilon RI signaling pathway | *MAP2K4, MAPK8, GAB2, VAV2* | *MAP2K4, MAPK8, GAB2, VAV2* |
| 4 | ssc04931: hsa04931 | Insulin resistance | *MLXIPL, MAPK8, PRKCE, CREB3L1, PTPRF* | *MLXIPL, MAPK8, CREB3L1, PRKCE, PTPRF* |
| 5 | ssc04140: hsa04140 | Regulation of autophagy | *ATG16L2, ATG7, ATG13* | *ATG16L2, ATG13, ATG7* |
| 6 | ssc04012 | ErbB signaling pathway | *CAMK2B, MAP2K4, MAPK8, CBLB* | |
| 7 | ssc04911 | Insulin secretion | *CAMK2B, CREB3L1, KCNMA1, RAPGEF4* | |
| 8 | hsa04520 | Adherens junction | | *GUCY1A2, KCNMA1, CLCA1, ITPR2* |
| 9 | hsa04514 | Cell adhesion molecules (CAMs) | | *CAMK2B, MAPK8, PRKCE, ITPR2, IL1RAP* |
| 10 | hsa04971 | Gastric acid secretion | | *PTPRM, PTPRJ, CTNNA3, CTNNA2, PTPRF* |
| 11 | hsa04724 | Glutamatergic synapse | | *GALNT14, GALNT13, GALNT10* |
| 12 | hsa04750 | Inflammatory mediator regulation of TRP channels | | *CNTNAP2, CDH2, PTPRM, CDH15, NEO1, PTPRF* |
| 13 | hsa00512 | Mucin type O-Glycan biosynthesis | | *CAMK2B, KCNK10, SLC26A7, ITPR2* |
| 14 | hsa04924 | Renin secretion | | *GRIN3A, GRM8, ITPR2, DLGAP1, SHANK2* |

molecular function, cellular component, and biological process. The most significant GO terms in the upregulated condition were as follows: regulation of cell shape (GO:0009360), platelet-derived growth factor receptor signaling pathway (GO:0048008), positive regulation of protein kinase B signaling (GO:0051897), antigen processing and presentation of endogenous peptide antigen via MHC class I (GO:0019885), and innate immune response (GO:0045087). The significant GO terms in the downregulation were translation (GO:0006412), cytoplasmic translation (GO:0002181), ribosomal small subunit assembly (GO:0000028) etc. Fig 2A and 2B show the biological processes. Likewise, extracellular exosome (GO:0070062), cytosol (GO:0005829), and cytoplasm (GO:0005737) were among the enriched terms in cellular component and GTPase activator activity (GO:0005096), non-membrane spanning protein tyrosine kinase activity (GO:0004715), and zinc ion binding (GO:0008270) were among the enriched terms in Molecular function (S2 Fig and S4 File).

## Discussion

A robust immune response to outside challenges could help in the survival of a biological entity, and the blood is an important component that has a key role in the development of the immune system. Blood circulates throughout the tissues, recognizes foreign bodies and subsequently acts through the T and B-cells [49–51]. The L type pig breed has been reported to have a better immune system over the ET type minipig breed, and hence, the emphasis was given to overcome this issue by development of a better ET type pig breed which can be used in various biomedical studies. Genomic selection methods have been successfully beneficial in various studies to understand the molecular mechanism involved in trait specific features and phenotypic characteristics [33]. These selection methods are based on a strong statistical foundation to predict an accurate gene selection and are widely used to improve the trait characteristics by understanding the mechanisms involved in adapting to the situation according to environmental changes and other factors for better survival [52–54]. Although, these methods have

enabled us to identify genes of interest from huge data in the form of positive selective sweep, a differentially expressed gene could help us in identifying potential markers and therefore have an imperative role in the development of better breed to prevail over the existing problems. Henceforth, we integrated the results obtained from the WGS with the RNA-seq data to identify potential genes involved in the development of the L type pig breed over the ET type.

The sweep genes among the L vs ET minipig variety exhibiting a positive signature were identified using XP-EHH and iHS and ZHp statistical test. The identified common genes expressed in the blood samples were limited to a total number of genes to 35. Among them, there were 7 (AIF1L, DNAH9, GABBR2, GRTP1, HBB, HECW1, and HTR4) differentially expressed (upregulated and downregulated) genes with log2FC $\geq \pm1.5$ and p.adj $\leq 0.05$ in the L type minipig. Among them,, the identified key genes HECT, C2 and WW Domain Containing E3 Ubiquitin Protein Ligase 1 (HECW1) also known as NEDD4-like ubiquitin protein ligase 1 (NEDL1) protein-coding gene identified as a positively selected gene with GO annotations related to this gene include a ubiquitin-protein ligase activity. They regulate the bone morphogenetic protein signaling pathway during embryonic development and bone remodeling [55, 56]. From a complex protein-protein interaction network, it has been identified as actively involved with the *transforming growth factor-beta* (TGFB) signaling pathway and directly interacts with SMAD family proteins responsible for regulating cell development and growth [57]. These SMAD family proteins have been strongly correlated with the immune response. that the SMAD pathway regulates the production of IgA by B cells, maintains the protective mucosal barrier, and promotes the balanced differentiation of CD4+ T cells into inflammatory T helper type and suppressive FOXP3+ T regulatory cells [58–60].

The identified selective sweep gene AIF1L has been identified as an important molecule that has an essential role in cell survival and is actively involved in proinflammatory activities of immune cells such as monocytes/macrophages and activated T lymphocytes [61, 62]. Besides this, DNAH9 gene also identified as a selectively sweep and a positively expressed differentially expressed gene is known to have a crucial role in the cytoplasmic movement of organelles, also known as cytoplasmic dyneins and the bending of cilia and flagella with the help of molecular motor axonemal dyneins [63]. These motors also enable the response to a broad array of signals including phosphorylation, Ca2+, redox changes, and mechanical activation [64, 65]. It is also reported that the respiratory tract is lined with cilia which keep inhaled dust, smog, and potentially harmful microorganisms from entering the lungs and overexpression of such key genes could help in the survival of eukaryotes at the cellular level by ejecting dust and foreign bodies entering the cells [66]; additionally, it creates the water currents necessary for respiration and circulation in sponges and coelenterates as well. Whereas, GRTP1 also known as Growth hormone-regulated TBC protein 1, is an upregulated gene in the L type found to be involved in a growth related function [67].

Moreover, the identified HTR4 (5-Hydroxytryptamine Receptor 4), a positively selected gene found on chromosome 2, is a Protein Coding gene that is a member of the family of serotonin receptors, which are G protein coupled receptors that stimulate cAMP production in response to serotonin. Serotonin stimulates monocytes and lymphocytes, which influence the secretion of cytokines and is also reported for utilizing functions in the innate and adaptive immunity [68, 69].

HBB, a selective sweep gene identified as three-fold down regulated in the L-type mini breed located on chromosome 9 on the *Sscrofa11.1* genome assembly is a Hemoglobin Subunit Beta protein coding gene. It is directly involved with the innate immune system and associated with important pathways in biological processes such as oxygen transport, receptor-mediated endocytosis, blood coagulation, etc. [70, 71]. Among the identified key genes, GABBR2 is one of the GPCR family proteins via γ-aminobutyric acid signaling pathway reported to have a key

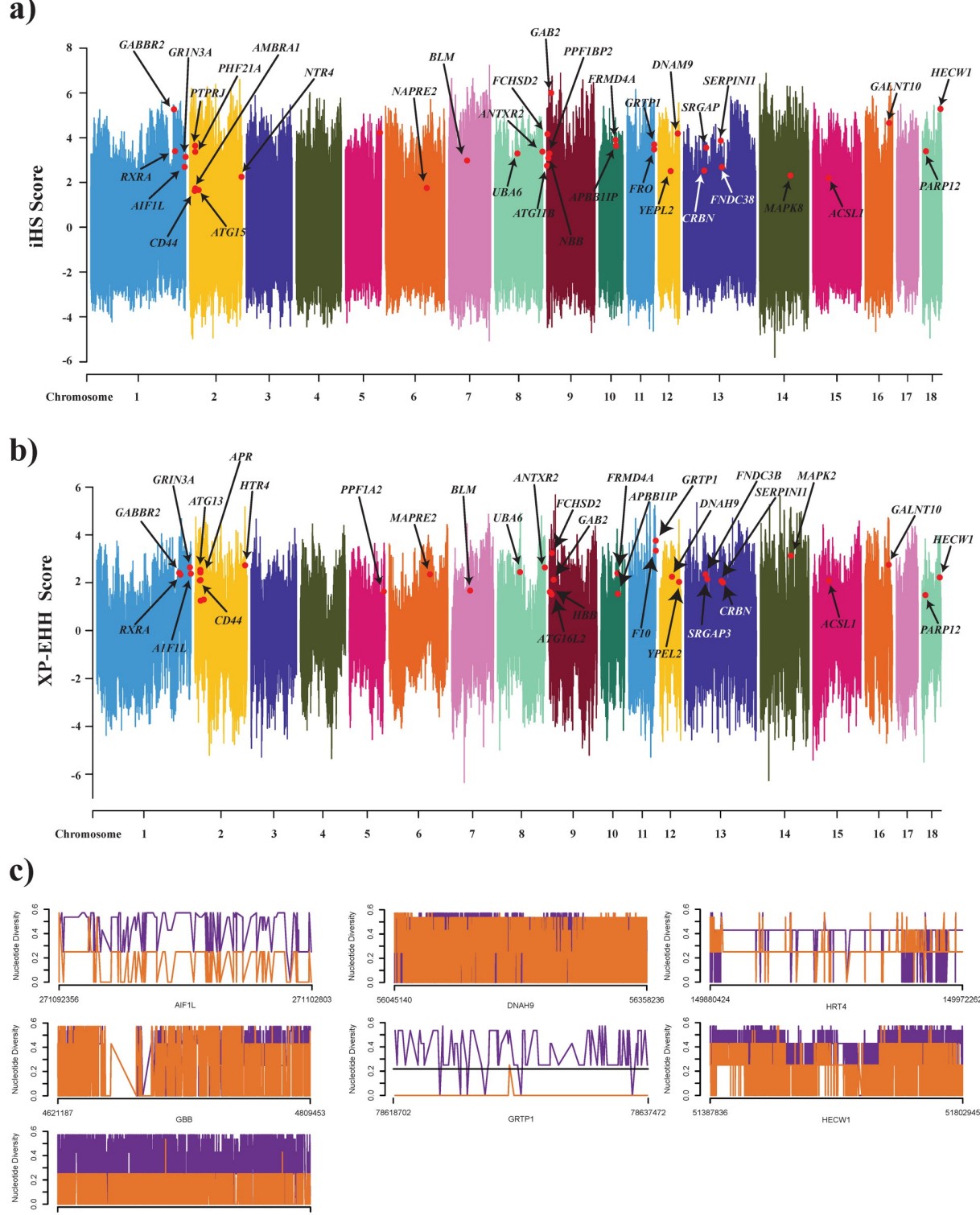

**Fig 3.** (a) Manhattan plot was generated to map the coordinates of identified genes on respective location of the chromosomes in iHS and XP-EHH. (b) These identified genes were further analyzed for nucleotide distribution comparison analysis in LvsET minipig and understanding the stability and gene level.

role in neurodevelopment phenotypes [72]. The characterization of this gene at the molecular level could help us to better understand how its downregulation helps the development of the L type minipig breed. Gene annotation studies also revealed a close association and active involvement of these genes in the various biological processes important in the immune system and in the development of cells at various stages (Fig 2A–2C). These identified genes were further mapped to their genomic positions using the Manhattan plot (Fig 3A and 3B). Afterwards, the identified genes were further analyzed for nucleotide diversity which also presented strong evidence of the stabilization effect in these genes in terms of selection (Fig 3C). In conclusion, we have identified 35 key genes among which 7 were differentially expressed and positively selected in the L-type Korean minipig. Nucleotide diversity analysis showed strong evidence for the stability of identified genes, and the gene ontology analysis revealed an association with the immune response associated pathways, regulation of autophagy, signaling pathways, etc. A comparative analysis with human states shows the importance of the minipig as a suitable animal model for various research. A comparative analysis was done to understand the similarity between the pathway association with humans as a reference source. The results clearly present evidence of a close association of different pathways at the molecular level and a strong association between them. The identified genes could be used as potential markers in molecular breeding processes and could enhance the immune response in the relative ET type minipig breed.

## Supporting information

**S1 File. Different tabs provided the statistical analysis result from iHS, XP-EHH, and ZGp score with differentially expressed genes with p.adj $\leq$ 0.05.**
(XLSX)

**S2 File. Common genes identified using statistical test integrated with DEGs.**
(XLSX)

**S3 File. Gene ontology analysis of selective sweep genes consist of (BP, CC, and MF) with background pig and human.**
(XLSX)

**S4 File. Gene ontology analysis of DEGs consist of (BP, CC, MF, and KEGG) with from upregulated and downregulated genes in L-type minipig.**
(XLSX)

**S1 Fig. Comparative KEGG pathway analysis of selective sweep genes with background pig and human to identify the commonly associated pathways using dot plot analysis.**
(TIF)

**S2 Fig. Gene ontology analysis viz cellular component and molecular function terms associated with upregulated and downregulated genes in L-type minipig was performed using REVIGO.**
(TIF)

## Author Contributions

**Conceptualization:** Devender Arora, In-Cheol Cho, Tae-Hun Kim, Woncheoul Park.

**Data curation:** Devender Arora, Kyung Soo Kang, Tae-Hun Kim, Woncheoul Park.

**Formal analysis:** Devender Arora, Jong-Eun Park, Dajeong Lim, Woncheoul Park.

**Funding acquisition:** Woncheoul Park.

**Investigation:** In-Cheol Cho, Woncheoul Park.

**Methodology:** Devender Arora, Woncheoul Park.

**Project administration:** Tae-Hun Kim, Woncheoul Park.

**Resources:** In-Cheol Cho, Tae-Hun Kim, Woncheoul Park.

**Software:** Devender Arora.

**Supervision:** Jong-Eun Park, Dajeong Lim, In-Cheol Cho, Woncheoul Park.

**Validation:** Woncheoul Park.

**Visualization:** Devender Arora, Jong-Eun Park, Dajeong Lim, Woncheoul Park.

**Writing – original draft:** Devender Arora.

**Writing – review & editing:** Devender Arora, Jong-Eun Park, Dajeong Lim, In-Cheol Cho, Kyung Soo Kang, Tae-Hun Kim, Woncheoul Park.

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
