## [Decision Letter · Decision Letter 0]

23 Nov 2021

PONE-D-21-34459Multi-omics Approaches for Comprehensive Analysis and Understanding of Immune Response in the Miniature Pig BreedPLOS ONE

Dear Dr. Park,

Thank you for submitting your manuscript to PLOS ONE. After careful consideration, we feel that it has merit but does not fully meet PLOS ONE’s publication criteria as it currently stands. Therefore, we invite you to submit a revised version of the manuscript that addresses the points raised during the review process.

ACADEMIC EDITOR: As suggested by both the reviewers the MS in its current from is full of grammatical and syntactical errors. Authors are instructed to edit the MS for English language with the help of either a native English speaker or a reputed English editing service provider (in both cases certificate/proof has to be attached while uploading the revised MS). In addition, provide figures of acceptable quality as suggested by one of the reviewers.  

We look forward to receiving your revised manuscript.

Kind regards,

Tushar Kanti Dutta, Ph.D.

Academic Editor

PLOS ONE

Journal Requirements:

2.  To comply with PLOS ONE submissions requirements, in your Methods section, please provide additional information on the animal research and ensure you have included details on (1) methods of sacrifice, (2) methods of anesthesia and/or analgesia, and (3) efforts to alleviate suffering.

(This work was supported by Korea Post-Genome Project (Project title: Deciphering the reference genome and the discovery of trait-associated genes in Nanchukmacdon and mini pigs). Project No. PJ013343 of the National Institute of Animal Science, Rural Development Administration, Republic of Korea. This study was supported by 2021 the RDA Fellowship Program of National Institute of Animal Science, Rural Development Administration, Republic of Korea. This funding helped in successfully performing all the sample analysis and provides the financial assistance to D.A.)

(This funding helped in successfully performing all the sample analysis and provides the financial assistance to D.A Korea Post-Genome Project (Project No. PJ013343) and RDA Fellowship Program of National Institute of Animal Science, Rural Development Administration, Republic of Korea. No: The funders had no role in study design, data collection and analysis, decision to publish, or preparation of the manuscript.)

Additional Editor Comments:

As suggested by both the reviewers the MS in its current from is full of grammatical and syntactical errors. Authors are instructed to edit the MS for English language with the help of either a native English speaker or a reputed English editing service provider (in both cases certificate/proof has to be attached while uploading the revised MS). In addition, provide figures of acceptable quality as suggested by one of the reviewers.

Reviewers' comments:

Reviewer's Responses to Questions

**Comments to the Author**

1. Is the manuscript technically sound, and do the data support the conclusions?

Reviewer #1: Yes

Reviewer #2: Yes

2. Has the statistical analysis been performed appropriately and rigorously? 

Reviewer #1: Yes

Reviewer #2: Yes

3. Have the authors made all data underlying the findings in their manuscript fully available?

Reviewer #1: Yes

Reviewer #2: Yes

4. Is the manuscript presented in an intelligible fashion and written in standard English?

Reviewer #1: No

Reviewer #2: No

5. Review Comments to the Author

Reviewer #1: General comments # 1: There are issues with the usage of English

General comments # 2: The text does not flow logically in the introduction, results and discussion, thereby greatly hampering the readability.

General comments # 3: The authors have not been careful and critical while drafting the manuscript. This is indicated by the spelling mistakes, undefined abbreviations and grammatical errors etc.

General comments # 4: All the figures should be replaced with good quality images

Reviewer #2: Article entitled, “ Multi-omics approaches for comprehensive analysis and understanding of immune response in the immature pig breed” is an unique study and is helpful for future approach for biomedical research. Therefore, I recommend this MS for publication after incorporating very minor corrections:

Line 44: Due to different size …..(the sentence needs correction). In my opinion this should be written as “ Due to different size and varied immune response, the test result is not mimicked in rodents which is otherwise possible in pig models. This has popularized pig for feasible use in biomedical research (6,7).

Line 49: Please delete the word dramatic

Line 50: “Furthermore, the immune system of minipig………preclinical trials.” Please make it two sentences.

Line 56: Why unnecessarily after genetic engineering “Hominidae” has been mentioned in bracket. Authors please justify.

The author is requested to correct the MS by an English speaking person specifically for Discussion and Result part.

6. PLOS authors have the option to publish the peer review history of their article (what does this mean?). If published, this will include your full peer review and any attached files.

Reviewer #1: **Yes: **Sadeesh E.M

Reviewer #2: No

---

## [Author Response · Author response to Decision Letter 0]

4 Jan 2022

Comment 1: Please ensure that your manuscript meets PLOS ONE's style requirements, including those for file naming.

Response: We are thankful to the editor for highlighting the formatting issue. We have made changes in the manuscript as per PlosOne standard.

Comment 2: To comply with PLOS ONE submissions requirements, in your Methods section, please provide additional information on the animal research and ensure you have included details on (1) methods of sacrifice, (2) methods of anesthesia and/or analgesia, and (3) efforts to alleviate suffering.

Response: Method section has been updated with the added information and provided the supporting reference used to perform the experiment.

Comment 3: We note that you have provided funding information that is not currently declared in your Funding Statement. However, funding information should not appear in the Acknowledgments section or other areas of your manuscript. We will only publish funding information present in the Funding Statement section of the online submission form. Please remove any funding-related text from the manuscript and let us know how you would like to update your Funding Statement.

Response: We have removed the funding statement from the main manuscript and provided the same with the cover letter.

Comment 4: Your ethics statement should only appear in the Methods section of your manuscript. If your ethics statement is written in any section besides the Methods, please move it to the Methods section and delete it from any other section. Please ensure that your ethics statement is included in your manuscript, as the ethics statement entered into the online submission form will not be published alongside your manuscript. 

Response: Done as suggested.

Comment 5: Additional Editor Comments:

As suggested by both the reviewers the MS in its current from is full of grammatical and syntactical errors. Authors are instructed to edit the MS for English language with the help of either a native English speaker or a reputed English editing service provider (in both cases certificate/proof has to be attached while uploading the revised MS). In addition, provide figures of acceptable quality as suggested by one of the reviewers.

Response: We have critically evaluated and used English editing service to improve grammatical errors in the manuscript.

Reviewer #1:

Comment 1: General comments # 1: There are issues with the usage of English

Response: We are thankful to the reviewer for highlighting the concern issue. We have taken this as critical issue and consulted with English expert to fix all the issue related with manuscript writing.

Comment 2: General comments # 2: The text does not flow logically in the introduction, results and discussion, thereby greatly hampering the readability.

Response: In the revised manuscript we have taken all precaution and re-write the mention section for easy to understand and maintaining the continuity of work flow.

Comment 3: The authors have not been careful and critical while drafting the manuscript. This is indicated by the spelling mistakes, undefined abbreviations and grammatical errors etc.

Response: Done as suggested.

Comment 4: General comments # 4: All the figures should be replaced with good quality images

Response: All the figure quality has been updated and replaced. 

Reviewer #2:

Article entitled, “ Multi-omics approaches for comprehensive analysis and understanding of immune response in the immature pig breed” is an unique study and is helpful for future approach for biomedical research. Therefore, I recommend this MS for publication after incorporating very minor corrections:

We are grateful to the reviewer for critically examine the manuscript and providing the positive feedback for the same. We have incorporated all the changes as suggested by the reviewer in the revised submission.

Comment 1: Line 44: Due to different size …..(the sentence needs correction). In my opinion this should be written as “ Due to different size and varied immune response, the test result is not mimicked in rodents which is otherwise possible in pig models. This has popularized pig for feasible use in biomedical research (6,7).

Response: Done as suggested.

Comment 2: Line 49: Please delete the word dramatic

Response: Done as suggested.

Comment 3: Line 50: “Furthermore, the immune system of minipig………preclinical trials.” Please make it two sentences.

Response: We re-wrote the mention section for clear and understanding.

Comment 4: Line 56: Why unnecessarily after genetic engineering “Hominidae” has been mentioned in bracket. Authors please justify.

Response: We agreed with the reviewer comment and remove the same from the manuscript.

---

## [Editor Report · Decision Letter 1]

11 Jan 2022

Multi-omics Approaches for Comprehensive Analysis and Understanding of the Immune Response in the Miniature Pig Breed

PONE-D-21-34459R1

Dear Dr. Park,

We’re pleased to inform you that your manuscript has been judged scientifically suitable for publication and will be formally accepted for publication once it meets all outstanding technical requirements.

Kind regards,

Tushar Kanti Dutta, Ph.D.

Academic Editor

PLOS ONE

Additional Editor Comments (optional):

I am provisionally accepting it to save time. However, at proof reading stage, kindly check figure legend details in page number 22. The information about S1 Fig and S2 Fig should go to supplementary file/data.
---

## [Editor Report · Acceptance letter]

10 May 2022

PONE-D-21-34459R1 

Multi-omics Approaches for Comprehensive Analysis and Understanding of the Immune Response in the Miniature Pig Breed 

Dear Dr. Park:

I'm pleased to inform you that your manuscript has been deemed suitable for publication in PLOS ONE. Congratulations! Your manuscript is now with our production department. 

Kind regards, 

on behalf of

Dr. Tushar Kanti Dutta 

Academic Editor

PLOS ONE